# Regulation of the Innate Immune System as a Therapeutic Approach to Supporting Respiratory Function in ALS

**DOI:** 10.3390/cells12071031

**Published:** 2023-03-28

**Authors:** Michael S. McGrath, Rongzhen Zhang, Paige M. Bracci, Ari Azhir, Bruce D. Forrest

**Affiliations:** 1Department of Medicine, University of California San Francisco, San Francisco, CA 94110, USA; 2Neuvivo, Inc., Palo Alto, CA 94301, USA; 3Department of Epidemiology and Biostatistics, University of California San Francisco, San Francisco, CA 94158, USA; 4Hudson Innovations, LLC, Nyack, NY 10960, USA

**Keywords:** ALS, VC, CRP, innate immunity, macrophage, inflammation, TGFB1, A2M, SAA, NP001

## Abstract

Amyotrophic lateral sclerosis (ALS) is a clinical diagnosis used to define a neurodegenerative process that involves progressive loss of voluntary muscle function and leads to death within 2–5 years after diagnosis, in most cases because of respiratory function failure. Respiratory vital capacity (VC) measurements are reproducible and strong predictors of survival. To understand the role of the innate immune response in progressive VC loss we evaluated ALS clinical trial and biomarker results from a 6-month phase 2 study of NP001, a regulator of innate immune function. All ALS baseline values were similar between treated and controls except for those > 65 years old who were excluded from analysis. Treated patients with plasma CRP ≥ 1.13 mg/L (high CRP) showed a 64% slower rate of VC decline compared with placebo and those with plasma CRP < 1.13 mg/L (low CRP) who showed no response. High CRP patients showed no age associated loss of VC whereas low CRP patients showed an age dependent loss of VC function. Plasma levels of serum amyloid A (SAA) were similarly elevated in high CRP patients consistent with ongoing innate immune activation. Plasma TGFB1 in high CRP treated patients was 95% higher than placebo at 6-months, confirming the activation and release of this anti-inflammatory factor by the innate immune alpha 2 macroglobulin (A2M) system. This report is the first to link a biomarker confirmed regulation of the innate immune system with a therapeutic approach for controlling VC loss in ALS patients.

## 1. Introduction

Amyotrophic lateral sclerosis (ALS) is a heterogeneous disease that involves gradual loss of motor neuron function leading to progressive paralysis and loss of respiratory function [1,2,3]. Loss of respiratory function is the major cause of death and the progressive loss as measured by standard vital capacity (VC) testing is predictive of survival [4]. To date no therapy either approved or under investigation has made any significant difference in rate of VC loss in ALS patients. 

Recent studies on ALS pathogenesis have identified initial events as occurring at the neuromuscular junction where neuron axonal processes interact with muscle outside the central nervous system [5,6,7]. This reaction is inflammatory and is mediated by components of the innate immune system including acute phase reactant proteins and blood derived granulocytes and macrophages. The triggering events have been linked to the presence of abnormally folded or aggregated proteins associated with ALS such as TDP43 and SOD-1, recognized by the innate immune system within the neuromuscular junction [8]. Normally, the acute phase reaction is followed closely by immune signals that turn off that reaction so as to have the immune system remain in balance overall [9,10]. Therefore, the plasma based, or humoral innate immune system serves to feedback on activated macrophages, the cellular drivers of inflammation, and balance the inflammatory with anti-inflammatory immune signals. Proteins associated with the acute phase innate immune response can be measured in the blood and one, C-reactive protein (CRP), allows quantitative determination of the degree of inflammation associated with disease [11].

Plasma CRP levels are acutely elevated more than 30-fold in an acute phase reaction, but rapidly fall as the initiator of the inflammation becomes controlled [12]. Two categories of insult cause the synthesis of CRP predominantly from the liver; infection and tissue damage [13]. Infections can trigger both the innate as well as the adaptive response so that blood levels of factors produced by both immune reactions are present in the plasma. Although CRP rapidly appears after an insult and is viewed as a marker for inflammation severity, its general function is to facilitate phagocytic cell clearance of foreign material. In addition, CRP downregulates other proinflammatory components of the innate immune response so as to turn down the production of inflammatory byproducts [14]. Therefore, if CRP levels are chronically elevated there should be ongoing infectious or tissue damaging processes driving the persistent CRP response, an acute phase process of the innate immune response. Importantly, continued tissue damage absent the persistence of CRP response suggests uncoupling of the innate immune response to signals that normally drive the anti-inflammatory reaction.

Studies that have confirmed the anti-inflammatory role for CRP in the immune response are many. CRP is known to induce myeloid derived suppressor cells that act to feedback on the proinflammatory component of the inflammation process [15]. CRP knockout animals have greater levels of autoimmune inflammation, atherosclerosis and liver disease progression [16,17,18]. The conventional use of plasma CRP levels to define cardiovascular risk is not defining CRP as an inflammatory driver of disease, but most likely as being elevated to control ongoing inflammatory activity [19,20].

In the context of ALS, CRP levels have been described as slightly elevated and higher in patients with more rapidly progressive disease [9,21]. Other studies have not confirmed CRP as a marker for ALS disease activity or survival [22]. To date, there is little consensus as to what CRP may be doing in the context of ALS pathogenesis.

Other plasma factors involved in the innate immune response include serum amyloid A (SAA), levels which increase in parallel with CRP [23]. SAA binds to and clears bacterial byproducts such as lipopolysaccharide (LPS). Alpha 2 macroglobulin (A2M) becomes activated when the acute phase reaction is initiated and serves to clear byproducts of damaged cells, specifically proteases [24]. When further activated by hypochlorite, a byproduct of the oxidative burst reaction initiated by phagocyte (granulocytes, macrophages) activation, A2M forms a dimer, releases preformed TGFB1 and binds to and removes misfolded proteins and aggregates. TGFB1 is a potent regulator of inflammation and serves to downregulate inflammatory drive [25]. Together, CRP, SAA and A2M represent major humoral components of the innate immune system and coordinate to regulate the degree of inflammatory reactions associated with the activated cellular components of the response (macrophages, granulocytes). The goal of the current study was to test whether the pathogenesis of ALS involved dysfunction of the innate immune system by using data generated from ALS clinical trials that involved a regulator of the innate immune system, NP001.

NP001, a formulation of sodium chlorite with known innate immune system regulatory activities, has been tested in three placebo controlled clinical trials in patients with ALS [26,27,28]. Biomarker linked anti-inflammatory activity was associated with positive trends in clinical outcomes in these trials. Considering the potential importance of CRP in regulating the innate immune response we reanalyzed data from the original NP001 phase 2A clinical trial [26] to determine whether CRP or other components of the innate immune system might be involved in the pathogenesis of ALS. The following study defines relationships between innate immune function and disease activity as defined by quantitative assessment of VC functional measures in NP001 treated, as compared to control ALS patients.

## 2. Materials and Methods

### 2.1. Description of ALS Phase 2A Trial and Participants

The NP001 phase 2A trial (ClinicalTrials.gov: NCT01281631, initiated on 24 January 2011) was conducted by Neuraltus Pharmaceuticals, Inc. (Palo Alto, CA, USA) from January 2011 to November 2012 at 17 sites in the United States. Details of this six-month, double-blind, placebo-controlled trial and analysis were published in 2015 [26]. Briefly, participants with symptom onset of less than 3 years and forced VC at baseline ≥ 70% of predicted value received a total of 20 infusions administered intravenously monthly over 6 cycles during a 6-month double-blind treatment period. Three groups including placebo, 1 mg/kg NP001 and 2 mg/kg NP001 were enrolled. There were 4 weeks between the start of each cycle. Cycle 1 consisted of 30 min infusions over 5 consecutive days. Cycles 2, 3, 4, 5 and 6 each consisted of 3 consecutive daily infusions. The Revised ALS Functional Rating Scale (ALSFRS-R) and vital capacity measurements were performed monthly. The ALSFRS-R, scored 0–48, was used to evaluate overall patient functional status [29,30]. The average rate of ALS disease progression (Average DP Rate) at baseline was defined as ALSFRS-R score change per month and calculated as follows:Average DP Rate = [(48 − ALSFRS-R score at baseline)/Months from ALS symptom onset to baseline].

Baseline high-sensitivity CRP (hs-CRP) measurements for all participants were evaluated and baseline plasma hs-CRP of 1.13 mg/L was used as a cutoff point for evaluating the role of inflammation in VC changes over the 6-month study [26,28]. The participants with plasma CRP ≥ 1.13 mg/L at baseline were defined as the high CRP group and the low CRP group included those with plasma CRP < 1.13 mg/L at baseline. For the purposes of this paper, we use the term CRP as a shortened version of high-sensitivity C-reactive protein. The units expressed throughout the paper are hs-CRP units, abbreviated as CRP.

The phase 2A clinical trial plasma specimens at baseline and at the end of the study were obtained from Neuvivo Inc. (Palo Alto, CA, USA), the biopharmaceutical company that obtained the materials from Neuraltus Pharmaceuticals, Inc. (Palo Alto, CA, USA) in 2019. The plasma specimens were stored at −80 ℃ since the trial ended in 2012. All patients within the 2 mg/kg NP001-treated and placebo arms who completed the 6-month study and had plasma available from baseline and at the end of the study (6-months) were included in the current biomarker study.

### 2.2. Analysis of Clinical Outcome Data

The clinical outcome data evaluated for the current study were the forced vital capacity measurements performed on all patients at baseline and every month until the end of study. Using the patient’s age and height with the actual respiratory volume measured, the % predicted vital capacity (VC) was calculated as described [26]. The clinical data analysis involved 125 individuals whose baseline CRP values were available from the phase 2A trial. Patients receiving 1 mg/kg NP001 showed an intermediate response compared to placebo and to the 2 mg/kg arm and thus were excluded from the current efficacy analysis. After the exclusion of patients > 65 years old and the group of patients randomized to receive 1 mg/kg NP001, 31 placebo (18 above CRP 1.13 mg/L, 13 below) and 30 NP001 2 mg/kg-treated (16 above CRP 1.13 mg/L, 14 below) participants were left for the efficacy and biomarker analyses. The distribution of participants enrolled in the NP001 phase 2A trial into categories whose data was analyzed in the current study is shown in Figure 1. 

### 2.3. Plasma Factors Evaluated

Acute phase reactant molecules, CRP and serum amyloid A (SAA) were measured directly and TGFB1 was measured as a surrogate marker for alpha 2 macroglobulin (A2M) activation into a dimeric form that releases preformed TGFB1 when stimulated by hypochlorite. Biomarker levels for specimens collected at baseline and end of study plasma were evaluated at the same time by the commercial company AssayGate, Inc. (Ijamsville, MD, USA). An extensive biomarker analysis was recently reported using specimens from this trial and baseline values of biomarkers fell within the expected range of those reported in the literature related to patients with ALS [31].

### 2.4. Statistical Analyses

Statistical analysis was performed using JMP Pro 16 (SAS Institute, Cary, NC, USA). Categorical data are summarized using counts and percentages and discrete and continuous factors are summarized using standard univariate descriptive statistics (number of participants, mean, standard deviation, median). Analysis of covariance models were used to compare the placebo to the NP001 group for continuous data. Prior to analysis, log transformations were applied to data that were not normally distributed. For all analyses, a two-sided *p*-value < 0.05 was considered statistically significant. 

## 3. Results

### 3.1. Plasma CRP Level Demographics

Figure 2 shows the baseline CRP distribution in the high and low CRP groups among the NP001 Phase 2A clinical trial participants. The median plasma CRP level was statistically significantly different between the groups (0.70 mg/L vs. 2.17 mg/L in low vs. high groups, respectively; *p* < 0.0001). Baseline CRP was not associated with baseline ALSFRS-R score or age.

### 3.2. Participants > 65 Years Old Have Significantly Different Relationships between Age and Times since Symptom Onset and Rates of Disease Progression

In previous studies of ALS patients enrolled in NP001 phase 2 trials those older than 65 years of age were shown to progress at different rates than those younger than 65 and were excluded from analysis in these reports [28,31]. Although CRP values were not associated with age, participants in the current analysis over the age of 65 differed significantly from patients 65 or younger in the disease activity variables of: (A) the rate of ALS disease progression (positively related to baseline values of CRP, Figure 3, R^2^ = 0.25, *p* = 0.04, *n* = 17); and (B) ALS disease duration (time from ALS symptom onset to baseline) (negatively related to baseline CRP values, Figure 4, R^2^ = 0.27, *p* = 0.03, *n* = 17). Because of the baseline disease activity associations with CRP in patients > 65, these patients were excluded from further analysis. 

### 3.3. Participants with Low Baseline CRP Show Age Dependent Loss of VC in Those ≤ 65 Years Old

Figure 5 shows a statistically significant negative linear relationship between age and baseline VC among patients with low baseline plasma CRP (R^2^ = 0.16, *p* = 0.007, *n* = 44). No linear relationship between age and baseline VC was observed for patients with high baseline plasma CRP (R^2^ = 0.00, *p* = 0.98, *n* = 62) (Figure 5).

### 3.4. Baseline Plasma CRP Values Define a Subset of NP001 Treated ALS Patients Whose Loss in VC over the 6-Month Phase 2A Study Is Markedly Slowed vs. Placebo

Further analyses were conducted to examine whether change in VC over time was associated with NP001 treatment. Figure 6 shows that loss of VC was slower over time in NP001 treated as compared to placebo in the high CRP group. However, for patients with low baseline CRP, change in VC over time did not differ by treatment status (Figure 7). Those patients with high CRP who were on placebo lost an average of 2.1% VC per month, whereas patients who were treated with NP001 lost 0.75% VC per month; a >64% slower rate of VC loss among those treated with NP001 compared with placebo (*p* = 0.05).

Baseline demographic and clinical characteristics among patents ≤ 65 years old with high CRP (Table 1) or ≤65 years old with low CRP (Table 2) were similar by treatment group. No significant differences were found between placebo and 2 mg/kg NP001 treated in both high and low CRP groups. 

### 3.5. NP001 May Function through Augmentation of the Normally Self-Regulated Innate Immune System

To test whether NP001 affects innate immune system function, factors related to innate immune system regulation were measured. Figure 8 shows the linear relationship between plasma levels of the acute phase reactant serum amyloid A (SAA) and CRP (R^2^ = 0.25, *p* = 0.004, *n* = 31) in baseline specimens from ALS patients with high CRP confirming that the immune activation in patients with high CRP is not isolated but a component of a generalized acute phase reaction.

Because CRP elevation as an immune regulator alone was not controlling ALS disease progression, we evaluated plasma TGFB1 to test whether there would be evidence for alpha 2 macroglobulin (A2M) activation, another major component of the humoral innate immune system [24], in patients responsive clinically to NP001. Figure 9 shows that TGFB1 levels increased significantly in treated patients with high baseline CRP as compared to placebo controls. The greater than 95% increase in plasma TGFB1 that occurred post exposure to NP001 vs. placebo, suggests that NP001 caused an A2M dimerization with release of TGFB1. The elevated level of TGFB1 one month after the last dose of NP001 also suggests that the other function of A2M dimers, the clearance of misfolded proteins, may have reset the innate immune activation cycle.

## 4. Discussion

One of the major challenges in developing an effective drug for the treatment of ALS is the identification of an ALS subset most likely to respond to the therapy. Neuvivo recently acquired results from three NP001 clinical trials performed in ALS patients by Neuraltus and as suggested by the 2019 FDA guidance [32] document evaluated the results and identified a patient population apparently responsive to NP001. 

The initial phase 2A study found that patients with plasma CRP levels above a median value for all of the randomized patients, 1.13 mg/L hs-CRP units, showed more of a clinical response to NP001 than those with CRP levels lower than 1.13 mg/L [26]. A recently published evaluation of the phase 2A study with additional biomarker analysis showed that baseline CRP values defined a subset of NP001 treated patients whose plasma levels of microbial translocation (MT) associated biomarkers, had disease slowed (ALSFRS-R loss) in conjunction with MT resolution [31]. Placebo controls failed to respond clinically and didn’t resolve any of the eight biomarkers that defined the outcome in the NP001 treated group. These data suggested that NP001 through regulation of the innate immune response could be benefiting those who showed evidence for innate immune system activity in the form of plasma CRP elevation.

The innate immune system responds rapidly to infection and/or tissue damage [13]. Within minutes, the acute phase reactants CRP and serum amyloid A (SAA) increase more than 30-fold in plasma [12]. CRP has been represented as a nonspecific marker for inflammation, however more than a marker, this molecule has innate immune functional activities. When an inflammatory response is initiated, byproducts of that response need to be removed or neutralized to avoid significant tissue damage. CRP facilitates the clearance of dead cells and protein aggregates. SAA binds to and removes products of bacterial clearance mediated by activated phagocytes such as LPS [23]. The role that CRP plays in other inflammatory diseases has been addressed in a variety of ways. CRP genetic knock out animal studies have shown that absent CRP, models of autoimmune arthritis, atherosclerosis, response to toxins, obesity and insulin resistance have all implicated CRP as an active regulator of the innate immune system rather than being a passive marker of inflammation [14,16,17,18,19,20]. A recent large study of Alzheimer’s disease and all forms of dementia linked low plasma CRP with greatest risk of disease [33]. The low CRP cohort of patients who showed an age-related loss of VC function may support the theory that an ongoing innate immune response as defined by persistence in CRP elevation may be stabilizing VC function in ALS patients.

Figure 10 shows a step-by-step schematic that demonstrates how the innate immune system self regulates normally and may be augmented by the addition of NP001. The innate immune activation cycle is a rapid and controlled multistep self-regulated process that occurs after exposure of the immune system to infections or tissue damage. In the case of ALS, the most likely initiator of innate immune system activation is the presence of misfolded and or aggregated proteins such as TDP43 or SOD1 in the neuromuscular junction [34,35]. Both CRP and SAA respond in parallel as in a normal acute phase response, but in a subset of ALS patients there is a persistence of these factors in the plasma (step 1). The innate inflammatory response is associated with an oxidative burst and the production of hypochlorous acid (HCLO) which activates alpha 2 macroglobulin (A2M) to bind and clear proteases (step 2). The normal oxidative burst byproduct HCLO is converted to taurine chloramine (TauCL) a regulator of NFkB and inflammation [36,37]. With excess HCLO, A2M dimerizes and releases preformed TGFB1 and binds to and clears misfolded proteins (step 3). TGFB1 feeds back on proinflammatory cells and turns off NFkB (step 4) [38]. The data presented in the current study are consistent with NP001 augmenting the innate immune system regulatory process after conversion in vivo to HCLO and stimulating steps 3 and 4 of the innate immune system regulation cycle (step 5) [39]. The greater than 95% increased plasma level of TGFB1 in ALS patients who received NP001 and show a vital capacity clinical response is consistent with NP001 augmenting an underperforming innate immune response.

In the current study, ALS patient VC was measured over time after initiation of NP001 treatment. High CRP patients showed a >64% slowing of VC loss as compared to high CRP placebos; low CRP patients showed no response to NP001. If CRP is an active component of an immune reaction that is being augmented by NP001, what is the innate immune system target? The potential answer is simple if the innate immune system is triggered in part by aggregated and misfolded proteins, both of which are features of neurodegenerative diseases [40]. In the context of ALS, the most likely target would be some form of TDP43 aggregate or misfolded protein present in the neuromuscular junction recognized by blood derived macrophages.

Respiratory vital capacity is linked directly to the function of the diaphragm innervated by the phrenic nerve and given the objective nature of the VC test, may allow the direct evaluation of innate immune activity as it either protects VC function or allows progressive destruction. Animal model studies have confirmed that ALS inflammatory damage begins outside the central nervous system (CNS) with blood derived macrophages reacting against misfolded or aggregated proteins such as SOD-1 and TDP43 in the neuromuscular junction [5,6,7]. VC phrenic nerve activity has also been linked to degree of systemic inflammation with the report that plasma IL-6 levels in ALS patients correlated with mean bilateral phrenic nerve motor evoked potential (CMAP) measurements [41]. Therefore, VC measurements define the integrity of the phrenic nerve function and degree of systemic inflammation, both of which are directly related to overall survival [4,42].

As a final point, if the process that we’ve described is what has improved respiratory function in ALS patients it may be possible through a pharmacodynamic approach to regulate the innate immune function to a point where no further damage would be forth coming in patients adequately controlled with NP001. As an extension to this possibility, other diseases that are affected by the innate immune system such as Alzheimer’s disease, Parkinson’s disease and neuropsychiatric disorders such as depression might be similarly regulated [43].

## 5. Limitations of the Current Study

The pathogenesis of ALS is still poorly understood but the linkage of respiratory function with survival is clear and any therapeutic approach that positively affects VC may extend survival in patients with ALS. The data presented in the current paper were derived from a phase 2 study of NP001 that was underpowered to define a true therapeutic potential for use of NP001 except in a subset of patients defined in post hoc studies [28,31]. Clearly, a larger properly powered placebo-controlled trial would need to be performed in order to test the theory that clinical VC responses to NP001 would be related to regulation of the innate immune system. The data presented in this and two previous papers analyzing clinical data from NP001 phase 2 trials [28,31] identified plasma CRP levels as critical to apparent clinical responsiveness to NP001. Given that CRP is an acute phase reactant, inherently variable plasma CRP levels that differ between patients can confound results tied to the selection of patients with slightly elevated CRP values. Others have had the same issues with CRP variability and a major source of the variability is related to age, specifically in those older than 65 [44]. In future studies a panel of innate immune system markers might allow a more definitive linkage between the pathogenesis of ALS and innate immune system regulation.

## Figures and Tables

**Figure 1 cells-12-01031-f001:**
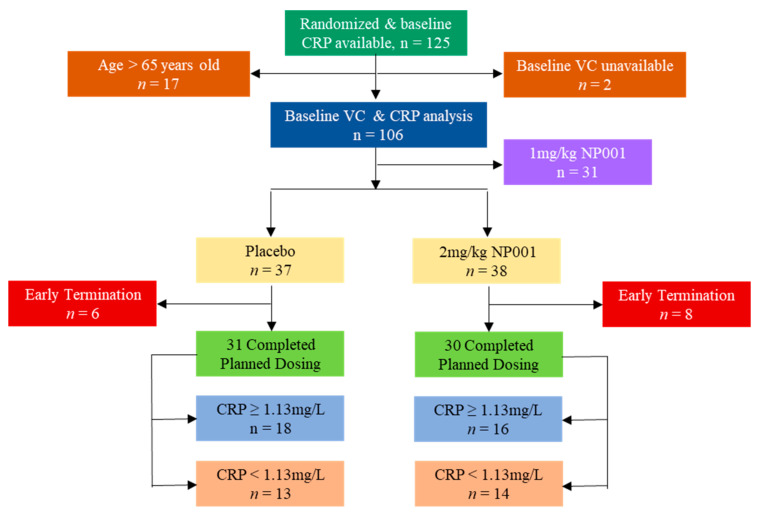
The flow chart of participants whose clinical data and specimens were included in the current study.

**Figure 2 cells-12-01031-f002:**
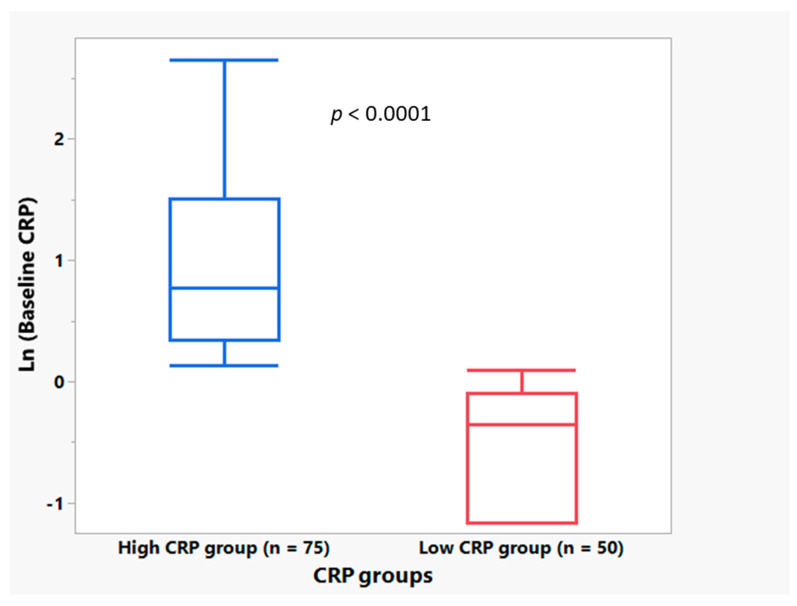
Box and Whisker plots depicting the distribution of baseline plasma CRP values for the high CRP group (CRP ≥ 1.13 mg/L) (*n* = 75 patients, in blue) and the low CRP group (CRP < 1.13 mg/L) (*n* = 50 patients, in red) where the bottom of the box depicts the first quartile of the data, the solid line through the box represents the median, and the top of the box depicts the third quartile of the data. The ends of the “whiskers” depict the maximum and minimum values. Results show that median baseline CRP value was statistically significantly higher in the high CRP group compared with the low CRP group (Wilcoxon test, *p* < 0.0001).

**Figure 3 cells-12-01031-f003:**
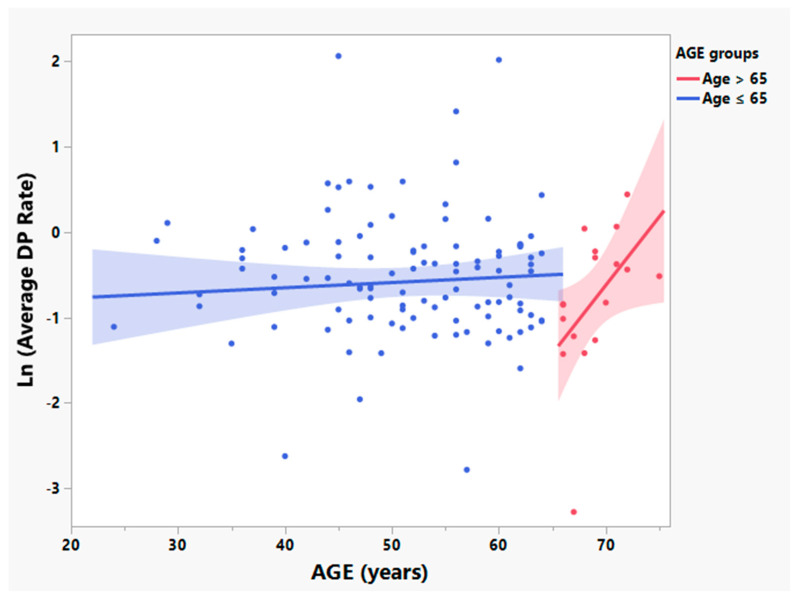
ALS patients aged > 65 years (depicted in red) showed baseline average rates of disease progression (Average DP Rate) different from those patients ≤ 65 years old in the phase 2A trial. Bivariate linear fit was used for the linear regression analysis, and showed a direct relationship between the log transformed DP rate (Average DP Rate) and age among participants > 65 years old (R^2^: 0.25, *p* = 0.04, *n* = 17, depicted in red). No linear association between age and Average DP Rate was found in those ≤ 65 years old (R^2^ = 0.00, *p* = 0.51, *n* = 108, depicted in blue).

**Figure 4 cells-12-01031-f004:**
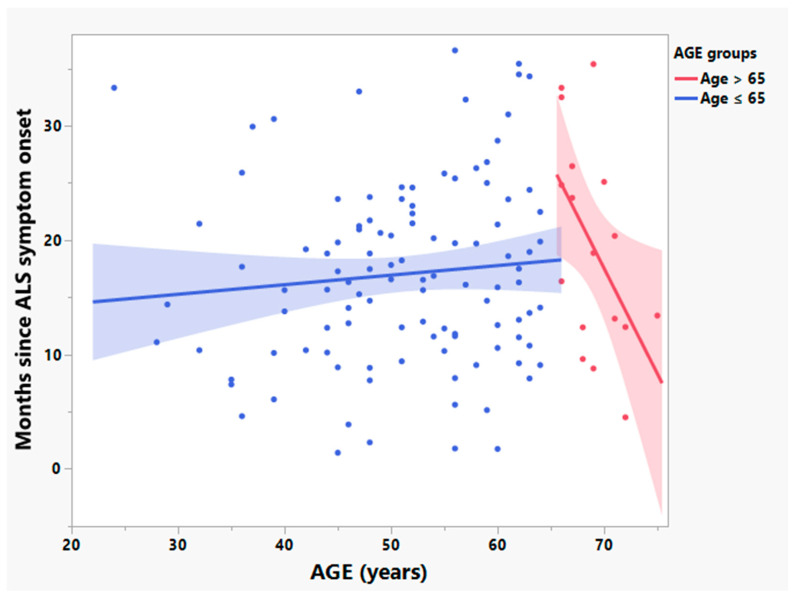
Among ALS patients in the phase 2A NP001 trial, duration of ALS symptom onset differed by age group (≤65 or >65 years old). Bivariate linear fit was used for the linear regression analysis. A negative relationship between the log transformed ALS disease duration (time since ALS symptom onset to baseline) and age was observed in the participants >65 years old (R^2^: 0.27, *p* = 0.03, *n* = 17, depicted in red). No linear association between age and ALS disease duration was found in those ≤ 65 years old (R^2^ = 0.01, *p* = 0.32, *n* = 108, depicted in blue).

**Figure 5 cells-12-01031-f005:**
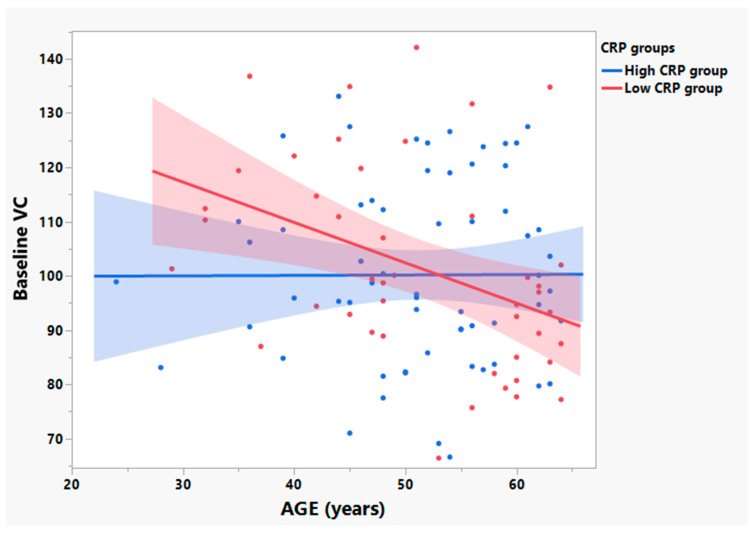
Baseline vital capacity (VC) values worsen with age in ALS patients with low baseline plasma CRP as compared to those with high plasma CRP in the participants age ≤ 65 years old. Bivariate linear fit was used for the linear regression analysis. In the participants aged ≤ 65 years old, baseline VC was negatively related to age in the low CRP group (R^2^ = 0.16, *p* = 0.007, *n* = 44, depicted in red). However, no linear association between baseline VC and age was observed in the high CRP group (R^2^ = 0.00, *p* = 0.98, *n* = 62, depicted in blue).

**Figure 6 cells-12-01031-f006:**
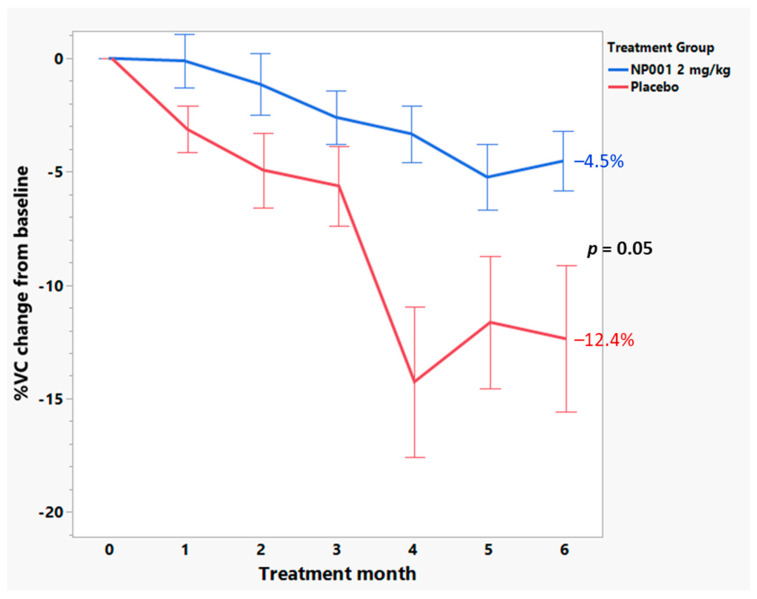
Among those aged ≤ 65 years with high baseline CRP, participants treated with NP001 experienced a slower vital capacity (VC) loss compared to placebo. %VC change from baseline for participants treated with NP001 (*n* = 16) is depicted in blue and is compared to the placebo group (*n* = 18) depicted in red. Error bars represent the range, mean ± SEM (standard error of the mean) of %VC change from baseline. The NP001 treatment group showed a 64% slower VC loss by the end of study (Wilcoxon test, *p* = 0.05).

**Figure 7 cells-12-01031-f007:**
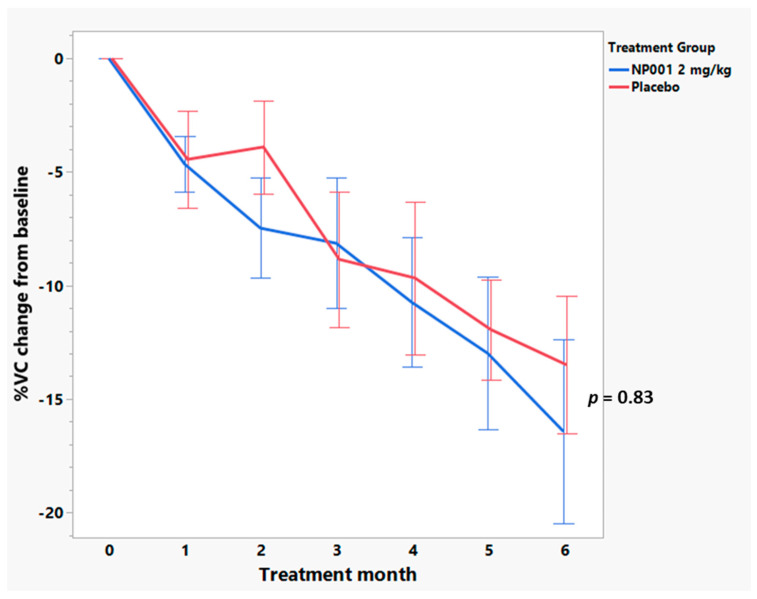
Comparison of vital capacity (VC) change from baseline over the six-month study in participants treated with NP001 compared to placebo in those aged ≤ 65 years old with low plasma CRP at baseline. %VC change from baseline for participants treated with NP001 (*n* = 14) depicted in blue and compared to placebo group (*n* = 13) depicted in red. Error bars represent the range, mean ± SEM (standard error of the mean) of %VC change from baseline. No differences were seen between NP001 and placebo groups by the end of study (NP001 = −16.4% vs. Placebo = −13.5%) (Wilcoxon test, *p* = 0.83).

**Figure 8 cells-12-01031-f008:**
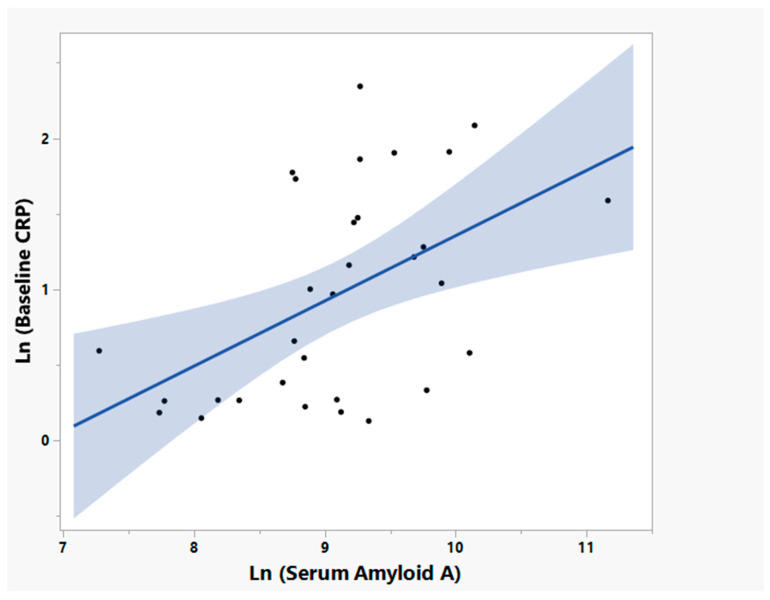
Positive linear relationship of the log transformed baseline plasma CRP and plasma Serum Amyloid A (SAA) in ALS patients age ≤ 65 years old and high plasma CRP at baseline (Bivariate linear regression analysis, R^2^ = 0.25, *p* = 0.004, *n* = 31).

**Figure 9 cells-12-01031-f009:**
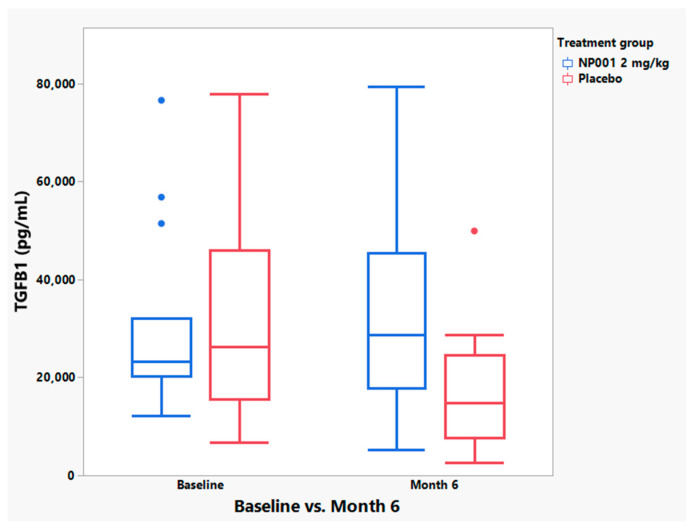
TGFB1 levels significantly increased over the 6-month trial in NP001 treatment compared to placebo controls in participants aged ≤ 65 years old with high plasma CRP at baseline. Box and Whisker plots depicting the distribution of baseline and end of study (month 6) plasma TGFB1 values for NP001-treated (*n* = 15 patients, in blue) and placebo group (*n* = 16 patients, in red) where the bottom of the box depicts the first quartile of the data, the solid line through the box represents the median, and the top of the box depicts the third quartile of the data. The ends of the “whiskers” depict the maximum and minimum values (Wilcoxon test, *p* = 0.89 for baseline and *p* = 0.02 for month 6).

**Figure 10 cells-12-01031-f010:**
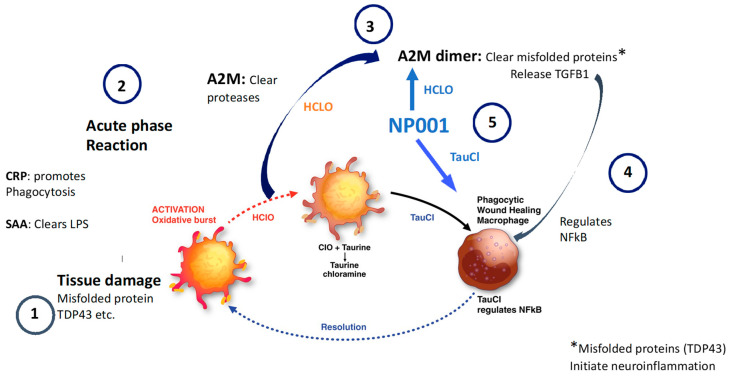
NP001 augments the innate immune activation cycle in ALS patients who have elevated baseline plasma CRP. Step 1: Blood derived macrophages become activated and begin clearance of cells and tissues containing pathogenic proteins and elaborate factors that initiate the acute phase reaction, producing hypochlorous acid as a byproduct of oxidative burst activation. Step 2: Rapid elevation of acute phase reactants including CRP and SAA, both of which amplify > 30×; CRP facilitates clearance of damaged tissues and SAA binds to and clears bacterial byproducts such as LPS. Alpha 2 macroglobulin becomes activated to absorb and remove damaged cell products such as proteases. Anti-inflammatory signaling begins. Step 3: Hypochlorous acid stimulates (1) production of the immune activation regulator taurine chloramine (TauCl). TauCl regulates macrophages to become phagocytic and wound healing and (2) A2M to dimerize and change function. A2M dimers clear misfolded proteins and release pre-synthesized TGFB1 to feedback on proinflammatory processes such as those driven by NFkB. Step 4: TGFB1 is a potent regulator of NFkB and inflammation. Macrophages exposed to TGFB1 become regulators of proinflammatory cells and the innate immune activation cycle is completed. Step 5: NP001 is converted by heme iron to hypochlorite which stimulates steps 3 and 4. TauCl regulates macrophages and hypochlorite stimulates further dimerization of A2M and release of TGFB1.

**Table 1 cells-12-01031-t001:** Baseline demographics and characteristics of patients ≤ 65 years old and plasma CRP ≥ 1.13 mg/L for those who completed the phase 2A study.

	NP001 2 mg/kg	Placebo	Overall
Characteristics	(*n* = 16)	(*n* = 18)	(*n* = 34)
Sex, *n* (%)			
Female	5 (31.3%)	5 (27.8%)	10 (29.4%)
Male	11 (68.8%)	13 (72.2%)	24 (70.6%)
Age at baseline, year	51.9 ± 9.9	51.8 ± 5.9	51.9 ± 7.9
Type of ALS, *n* (%)			
Familial	0 (0.0%)	2 (11.1%)	2 (5.9%)
Sporadic	16 (100.0%)	16 (88.9%)	32 (94.1%)
Site of ALS onset, *n* (%)			
Bulbar	3 (18.8%)	3 (16.7%)	6 (17.6%)
Limb	13 (81.3%)	15 (83.3%)	28 (82.4%)
El Escorial criteria for ALS, *n* (%)			
Definite	7 (43.8%)	9 (50.0%)	16 (47.1%)
Probable	8 (50.0%)	7 (38.9%)	15 (44.1%)
Possible	1 (6.3%)	2 (11.1%)	3 (8.8%)
Concurrent riluzole use, *n* (%)			
Yes	11 (68.8%)	12 (66.7%)	23 (67.6%)
No	5 (31.3%)	6 (33.3.%)	11 (32.4%)
ALSFRS-R score at baseline ^1^, Mean ± SD	37.7 ± 4.1	39.2 ± 3.9	38.5 ± 4.0
Vital capacity at baseline, Mean ± SD	113.5 ± 17.0	109.2 ± 15.3	111.2 ± 16.0
Months since ALS symptom onset ^2^, Mean ± SD	19.29 ± 7.63	13.76 ± 6.55	16.36 ± 7.52
CRP at Baseline (mg/L) ^3^, Mean ± SD	3.48 ± 2.57	2.95 ± 2.24	3.20 ± 2.38

Abbreviation: n, number of participants. SD, Standard Deviation. ^1^ ALSFRS-R score: The revised Amyotrophic lateral sclerosis functional rating scale. ^2^ Months from ALS symptom onset to baseline. ^3^ Baseline plasma levels of C-Reactive Protein.

**Table 2 cells-12-01031-t002:** Baseline demographics and characteristics of patients ≤ 65 years old and plasma CRP < 1.13 mg/L for those who completed the phase 2A study.

	NP001 2 mg/kg	Placebo	Overall
Characteristics	(*n* = 14)	(*n* = 13)	(*n* = 27)
Sex, *n* (%)			
Female	5 (35.7%)	4 (30.8%)	9 (33.3%)
Male	9 (64.3%)	9 (69.2%)	18 (66.7%)
Age at baseline, year	51.3 ± 8.9	51.8 ± 11.6	51.6 ± 10.1
Type of ALS, *n* (%)			
Familial	1 (7.1%)	2 (15.4%)	3 (11.1%)
Sporadic	13 (92.9%)	11 (84.6%)	24 (88.9%)
Site of ALS onset, *n* (%)			
Bulbar	2 (14.3%)	2 (15.4%)	4 (14.8%)
Limb	12 (85.7%)	11 (84.6%)	23 (85.2%)
El Escorial criteria for ALS, *n* (%)			
Definite	7 (50.0%)	7 (53.8%)	14 (51.9%)
Probable	7 (50.0%)	6 (46.2%)	13 (48.1%)
Concurrent riluzole use, *n* (%)			
Yes	10 (71.4%)	9 (69.2%)	19 (70.4%)
No	4 (28.6%)	4 (30.8%)	8 (29.6%)
ALSFRS-R score at baseline ^1^, Mean ± SD	38.4 ± 6.5	40.0 ± 3.2	39.2 ± 5.1
Vital capacity at baseline, Mean ± SD	110.3 ± 15.7	110.8 ± 19.0	110.6 ± 17.1
Months since ALS symptom onset ^2^, Mean ± SD	13.60 ± 7.18	16.56 ± 7.68	15.02 ± 7.43
CRP at Baseline (mg/L) ^3^, Mean ± SD	0.73 ± 0.26	0.62 ± 0.30	0.68 ± 0.28

Abbreviation: n, number of participants. SD, Standard Deviation. ^1^ ALSFRS-R score: The revised Amyotrophic lateral sclerosis functional rating scale. ^2^ Months from ALS symptom onset to baseline. ^3^ Baseline plasma levels of C-Reactive Protein.

## Data Availability

The data are available through Neuvivo, Inc. upon request.

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
