# Peer review of "Regulation of the Innate Immune System as a Therapeutic Approach to Supporting Respiratory Function in ALS"

_cells, 2023, doi:10.3390/cells12071031_

Round 1

Reviewer 1 Report

 This report gives interesting information about inflammatory changes and respiratory decline in ALS, and the possible importance of anti-inflammatory drugs in changing this interaction. This association has been explored before (see doi.org/10.1016/j.jns.2019.01.026).

Some points deserve more attention

Are the biomarkers stable in blood samples after 10 years even kept at -80ºC?

How were the CRP and disease duration time cutoffs defined? (Median-split?)

The authors concluded that patients over 65 had a more aggressive disease with faster VC change. Probably this derived from a larger percentage of patients with bulbar-onset disease. In this way, it would be better to use a statistical model to define the independent predictors of VC change (like as region of onset, age, disease duration and treatment) rather than exploring many direct comparisons.

Number the patients is quite small, not sure how many were analyzed after removing the older group. This number limitation should be stressed in discussion. Indeed in figure 8 we can observe a bizarre variation of VC in the untreated group that represents the risk if evaluating small groups.

We are not informed if TGFB1 changes overtime in the group of patients with lower CRP page 11)

There are too many figures, but the most relevant figure, younger patients with high CRP treated vs untreated, seems absent.

The wording VC “improved” in the treated group (results, discussion, abstract) is misleading, there was no improvement but a lower decline.

Results, point 3.4, should be moved to discussion. Discussion should not repeat introduction, should be shorter.

In table 1 and 2, the column overall is not necessary but the p values are missing.

Minor

Why placebos (plural) in the abstract?

Author Response

Response to reviewer #1’s comments:

  1. This report gives interesting information about inflammatory changes and respiratory decline in ALS, and the possible importance of anti-inflammatory drugs in changing this interaction. This association has been explored before (see doi.org/10.1016/j.jns.2019.01.026).

We have incorporated the findings from this publication that linked degree of inflammation as represented by serum IL-6 levels with phrenic nerve function in our discussion (reference 41). We also propose to use measures such as provided in this publication to assess respiratory function with drug activities in future clinical trials.

  1. Some points deserve more attention

  • Are the biomarkers stable in blood samples after 10 years even kept at -80ºC?

Plasma CRP values were obtained at the initiation of the clinical trial. A series of plasma biomarkers were evaluated in 2022 using drug treated and placebo derived patient specimens and values were reported in Zhang et. al. (reference 31). The biomarkers reported in the current manuscript were evaluated in the same time frame as reported in Zhang et. al. Actual quantitative values for all of the markers were reported and fell within ranges reported associated with ALS patients in the literature.

  • How were the CRP and disease duration time cutoffs defined? (Median-split?)

  1. In the first NP001 phase 2 trial reported in 2015 (Miller et. al. reference 26) plasma CRP levels were measured and in a prespecified manner the median hs-CRP value of everyone in the study of 1.13 mg/L (converted from wr-CRP) was used to define ALS patients as either more or less inflammatory.
  2. Disease duration, or time since symptom onset was data collected in the phase 2A NP001 trial. These data are reported in tables 1 and 2 along with other demographic information. We have removed Figures 7 and 8, no longer dividing patient groups into disease duration subgroups.

  • The authors concluded that patients over 65 had a more aggressive disease with faster VC change. Probably this derived from a larger percentage of patients with bulbar-onset disease. In this way, it would be better to use a statistical model to define the independent predictors of VC change (like as region of onset, age, disease duration and treatment) rather than exploring many direct comparisons.

  1. We have emphasized earlier publications that defined the rationale for separating patients above and below the age of 65 in the revised manuscript. For comparison across studies (references 28 & 31) we have used the same 65-year-old cutoff rationale and specify this in the revised manuscript.
  2. We have included a flow chart that shows the patient selection process used in the analysis.

  • Number the patients is quite small, not sure how many were analyzed after removing the older group. This number limitation should be stressed in discussion.

  1. The new Figure 1 is a flow chart showing the accounting of all patients enrolled in the phase 2A NP001 trial.
  2. We have added a limitations of study section at the end of the discussion.

  • Indeed, in figure 8 we can observe a bizarre variation of VC in the untreated group that represents the risk if evaluating small groups.

We have removed figures 7 and 8 as they represented subsets of an already small study.

  • We are not informed if TGFB1 changes overtime in the group of patients with lower CRP page 11)

  1. As noted in the new figure 10, now moved into the discussion section, the topic of the current study was the evaluation of the innate immune response as it was happening in ALS patients with CRP levels associated with a clinical improvement; those with CRP > 1.13 mg/L.
  2. In a previous publication (reference 31) evaluating ALS disease activity associated biomarkers, only those from patients with CRP > 1.13 showed any changes over the six-month trial in either the NP001 treated or placebo groups. Given specimen limitations, TGFB1 levels were not measured in the CRP< 1.13 group.

  • There are too many figures, but the most relevant figure, younger patients with high CRP treated vs untreated, seems absent.

Figures 5 and 6 represent the VC analyses of the young subset with high and low CRP respectively.

  • The wording VC “improved” in the treated group (results, discussion, abstract) is misleading, there was no improvement but a lower decline.

All notations suggesting improvement have been converted to slowed loss of VC function.

  • Results, point 3.4, should be moved to discussion. Discussion should not repeat introduction, should be shorter.
  1. The entire section related to the innate immune system model and the discussion related to the presented model have been removed or compressed. The discussion has been compressed.
  2. A final element related to Limitations of the study has been added at the end of the discussion.
  • In table 1 and 2, the column overall is not necessary but the p values are missing.

A sentence “No significant differences were found between placebo and 2mg/kg NP001 treated in both high and low CRP groups.” was added to the results section 3.4. in the revised version rather than show insignificant p values for Table 1 & Table 2.

  • Why placebos (plural) in the abstract?

Corrected.

Reviewer 2 Report

Your study provides evidence for a new approach to treating ALS with NP001 that targets the innate immune system, potentially improving respiratory function and ultimately prolonging survival. This is an important avenue and the innate immune system in ALS has not been previously recognized for treatments. Your study highlights the importance of early intervention, particularly in patients with high levels of inflammation. Overall, the findings suggest that regulation of the innate immune system may be a promising avenue for developing new ALS therapies.

However, it should be noted that this was a phase 2 clinical trial, and larger, randomized, double-blind, placebo-controlled trials are needed to confirm the results and establish the safety and efficacy of the treatment.

A succinct discussion of the limitations of the trial and possible avenues for future improvements are needed in the manuscript. Please discuss those.

In addition, there are serious limitations when we treat innate immune systems because of inherent variability between patients. Please discuss these in the revised manuscript.

Author Response

Response to reviewer #2’s comments:

Your study provides evidence for a new approach to treating ALS with NP001 that targets the innate immune system, potentially improving respiratory function and ultimately prolonging survival. This is an important avenue and the innate immune system in ALS has not been previously recognized for treatments. Your study highlights the importance of early intervention, particularly in patients with high levels of inflammation. Overall, the findings suggest that regulation of the innate immune system may be a promising avenue for developing new ALS therapies.

  • However, it should be noted that this was a phase 2 clinical trial, and larger, randomized, double-blind, placebo-controlled trials are needed to confirm the results and establish the safety and efficacy of the treatment.

We have added a limitation of study at the end of the discussion. Earlier reports referenced in this manuscript report on the efficacy and safety of NP001.

  • A succinct discussion of the limitations of the trial and possible avenues for future improvements are needed in the manuscript. Please discuss those.

The limitations of study section has been added at the end of the discussion section.

  • In addition, there are serious limitations when we treat innate immune systems because of inherent variability between patients. Please discuss these in the revised manuscript.

We have expanded discussion related to the variability of evaluating innate immune system activities in the context of further clinical studies.

Round 2

Reviewer 1 Report

This version is improved and this reviewer´s points were addressed

The sentences in lines 60-63 are not clear and should be corrected

Author Response

Response to reviewer #1’s second comments

  1. The sentences in lines 60-63 are not clear and should be corrected.

As the reviewer suggested, we have made changes to the contents in lines 60-63 as follows in the revised manuscript:

“Therefore, if CRP levels are chronically elevated there should be ongoing infectious or tissue damaging processes driving the persistent CRP response, an acute phase process of the innate immune response. Importantly, continued tissue damage absent the persistence of CRP response suggests uncoupling of the innate immune response to signals that normally drive the anti-inflammatory reaction.”
